# Chemical Characterization, Antioxidant, Cytotoxic and Microbiological Activities of the Essential Oil of Leaf of *Tithonia Diversifolia* (Hemsl) A. Gray (Asteraceae)

**DOI:** 10.3390/ph12010034

**Published:** 2019-03-04

**Authors:** Ana Luzia Ferreira Farias, Alex Bruno Lobato Rodrigues, Rosany Lopes Martins, Érica de Menezes Rabelo, Carlos Wagner Ferreira Farias, Sheylla Susan Moreira da Silva de Almeida

**Affiliations:** Laboratory of Pharmacognosy and Phytochemistry-Federal University of Amapá-Highway Jucelino Kubistichek, Km-02. Macapá, 68.902-280 Amapá, Brazil; analuziafarias@yahoo.com.br (A.L.F.F.); alexrodrigues.quim@gmail.com (A.B.L.R.); rosyufpa@gmail.com (R.L.M.); ericamrabelo@gmail.com (É.d.M.R.); carloswagnerfarias@gmail.com (C.W.F.F.)

**Keywords:** secondary metabolites, margaridão, medicinal plants

## Abstract

The present study aimed to evaluate the chemical composition, antioxidant potential, and the cytotoxic and antimicrobial activity of the essential oil of the plant species *Tithonia diversifolia* (Hemsl) A. Gray. The essential oil obtained was used to identify the chemical compounds present through the techniques of GC-MS and NMR. The antioxidant potential was calculated by the sequestration method of 2,2-diphenyl-1-picrylhydrazyl. For cytotoxic activity, the larval mortality of *Artemia salina* was evaluated. The main chemical constituents identified are *α*pinene (9.9%), Limonene (5.40%), *(Z)*-*β*-ocimene (4.02%), *p*-cymen-8-ol (3.0%), Piperitone (11.72%), *(E)*-nerolidol (3.78%) and Spathulenol (10.8%). In the evaluation of the antimicrobial activity, bacterial strains of *Staphylococcus aureus*, *Escherichia coli* and *Pseudomonas aeruginosa* were used. The results showed that the bacterium *E. coli* were more susceptible to the presence of the essential oil, presenting minimal inhibitory concentration at the concentrations that were exposed. The essential oil presented antioxidant activity of 54.6% at the concentration of 5 mg·mL^−1^ and provided a CI_50_ of 4.30. It was observed that the essential oil of this species was highly toxic against *A. salina* lavas, as its cytotoxic activity showed an LC_50_ of 3.11. Thus, it is concluded that *T. diversifolia* oils are effective in inhibiting bacterial growth and reducing oxidative stress.

## 1. Introduction

The *Asteraceae* (*Compositae*) family is known for its therapeutic, cosmetic and aromatic properties. Its main uses are anthelmintic, anti-inflammatory, astringent, cholesteric, anti-hemorrhagic, antimicrobial, antioxidant, diuretics, analgesics and antispasmodics [1,2,3,4,5].

Among the species of this family stands the *Tithonia diversifolia*, popularly known as Margaridão, the Amazonian flower, and the Mexican sunflower. Although this species is native to Mexico, it can already be found in different regions of Brazil and Africa [6].

In West Africa, leaf alcohol extracts are widely used for the treatment of chronic malaria, an alternative utilized by the low-income population that suffers from a high mortality rate of this disease, especially among children [7].

Many studies carried out to prove the biological activities of this species are limited in studies with crude extracts, mostly ethanolic extracts. Little is known about the biological activities that the essential oils of *T. diversifolia* can provide for the treatment of different diseases described by ethnopharmacology [8] such as constipation, stomach pains, indigestion, sore throat, liver pain, menstrual pain [9], anti-inflammatory activities [10], anticancer activity [11], anti-amoebic [12], antiviral and activity against the human immunodeficiency virus [13].

In addition, there is little research in Brazil that proves the biological activities of the essential oil of this species. In relation to the state of Amapá, which is considered the most conserved territory of the Brazilian Amazon [14], the results obtained in this work are unpublished. Thus, the objective of this study was to evaluate the chemical composition, antioxidant, cytotoxic and microbiological activity of the essential oil of the leaves of the species *T. diversifolia* (*Asteraceae*).

## 2. Results

### 2.1. Chemical Characterization of Essential Oil

The essential oil yield of *T. diversifolia* leaves, obtained by hydrodistillation, was 0.65% (m/m). The chemical composition can be observed in Table 1. The main constituents (Appendix A) are *α*-pinene (9.9%), Limonene (5.40%), *(Z)-β*-ocimene (4.02%), *p*-cymen-8-ol (3.0%), Piperitone (11.72%), *(E)*-nerolidol (3.78%) and Spathulenol (10.8%). 

For the confirmation of the major substances, *T. diversifolia* essential oil was analyzed by 1H and 13C NMR (Appendix A) described in Table 2, Table 3, Table 4, Table 5, Table 6, Table 7, Table 8 and Table 9.

The 1 H-NMR spectrum (Appendix A) was absorbed from the essential oil of *T. diversifolia*, this being a mixture of compounds, where differences in the intensities of the overlap signals can be observed.

### 2.2. Antioxidant Activity

The values obtained of the antioxidant activity %AA of the essential oil of the species *T. diversifolia* can be observed in Table 10: The values obtained of the antioxidant activity %AA of the essential oil of the species *T. diversifolia* can be observed in Table 10:

Horizontally, % AA values followed by the same letter did not present significant differences for ANOVA (*p* < 0.05).

The essential oil extracted from the leaves of *T. diversifolia* showed significant antioxidant activity and significant value with *p* < 0.0001 at the concentration of 5 mg·mL^−1^ with %AA 54.6 ± 0.06. The minimum inhibitory concentration (IC_50_) was 4.30 mg·mL^−1^, with a strong correlation coefficient (R^2^) of 0.9965. 

To evaluate the antioxidant potential the essential oil of *T. diversifolia* was used as an antioxidant standard constituent ascorbic acid. To achieve this, an assay was performed using the 2,2-diphenyl-1-picrylhydrazyl (DPPH)free radical consumption method (Table 11).

The values obtained indicated that compared with this standard (ascorbic acid), the essential oil of *T. diversifolia* had low antioxidant activity. The IC50 of ascorbic acid had an IC 50 of 16.71 μg·mL^−1^, a much lower value than that of essential oil, which was 4.30 mg·mL^−1^.

### 2.3. Cytotoxicity Test

Table 12 shows the readings performed and the calculations of mean mortality performed within 48 h of the cytotoxic activity of *T. diversifolia* essential oil against nauplii (*A. salina* larvae). The results are significant and are expressed as percent mortality (%).

Values with equal superscript letters represent significant equality in the concentrations of essential oils.

The essential oil of *T. diversifolia* presented LC_50_ of 3.11 μg·mL^−1^ and correlation coefficient of R2 0.999, evidencing high toxicity against larvae of *A. salina*.

### 2.4. Antimicrobial Activity

Table 13 shows the results corresponding to the minimum concentrations for inhibition of the growth of bacteria *Staphylococcus aureus*, *Escherichia coli* and *Pseudomonas aeruginosa*.

## 3. Discussion

### 3.1. Chemical Characterization of Essential Oil

The results of the chemical composition of the essential oil of leaves of the vegetal species of this study corroborate with the results of the research done by Lawal et al. [24], which is similar in composition to that found in this study.

### 3.2. Antioxidant Activity

The antioxidant activity was performed by the DPPH radical capture method (or DPPH consumption). This methodology consists of absorbance reading through a Spectrophotometer, the higher the absorbance values, the greater the antioxidant activity. The determination of the antioxidant activity was carried out using the DPPH free radical capture method, however, it is known that the essential oils are complex mixtures and it is difficult to attribute certain biological activities. In the present assay, the elimination of the radical was shown to be low. The aspect presented was still slightly purple. The antioxidant activity is directly related to the free radical consumption. The higher this consumption, the greater the activity and the lower the inhibitory concentration, in this case the IC 50 [25,26,27].

The antioxidant activity is related to the ability of a substance to eliminate free radicals; the radicals with this arrangement have unpaired electrons in their last electronic layer, making the compound highly unstable and reactive. Substances that have the ability to reduce oxidative stress caused by excessive oxidation can protect a biological system from different pathologies such as diabetes mellitus, multiple sclerosis, heart disease, Parkinson’s disease, inflammation, Alzheimer’s disease, atherosclerosis, stroke, and cancer [28].

According to Oliveira [29], the method of reducing or neutralizing the stable free radical, DPPH, can assess the antioxidant capacity of various substances. DPPH is widely used for this purpose due to its efficiency, practicality, and speed.

The presence of spathulenol (10.8%) as one of the major compounds of *T. diversifolia* essential oil may explain the antioxidant activity observed in this study. This result corroborates with research by Ćavar et al. [30] that verified a high presence of spathulenol and attributed this to the significant antioxidant activity verified in the oils of the species *Satureja montana* L. 

Researchers conducted by Roberto et al. [31] and Seol [32] also observed the antioxidant activity of limonene and linalool compounds, noting that they act as regulators of oxidative stress. It is observed the presence of these substances in the chemical composition of the essential oil of *T. diversifolia* (limonene 5.40% and linalool 0.32%), which could justify the results obtained for this activity.

### 3.3. Cytotoxicity Test

The larvae of *A. salina* (nauplii), are effective to verify if a certain plant sample is toxic. A preliminary toxicity test on plant species is important to verify that the study material may contain substances that cause damage to the health of a population. In addition, toxicity results may point to the possibility that the test material (*T. diversifolia* essential oil) may be used for biocidal, repellent and antitumor activities [33,34,35].

Studies performed by Silva et al. [36] and Eltayeib and Ishag [37], classify the toxicity of the plant as follows: values less than 100 μg·mL^−1^ are considered highly toxic, above 100 μg·mL^−1^ the classification considers them to be moderate, values greater than 500 μg·mL^−1^ are of low toxicity, and values with LD_50_ above 1000 μg·mL^−1^ are classified as non-toxic. In this way, the *T. diversifolia* oil can be classified as highly toxic. 

According to Riani et al. [38], the presence of piperitone in the chemical constitution of a species can make it toxic. In the same way, supporting the results obtained in this study, since the present chemical constituent is one of the main identified compounds of the species of study.

The World Health Organization conceptualizes medicinal plants as being species that have in some part of their organism the presence of chemical compounds that can be used for therapeutic purposes. In this sense, it is imperative that studies are conducted to evaluate the benefits that herbal medicines can provide. However, it is observed that certain species of plants have in their chemical composition substances that can be considered potentially harmful when not used correctly [39]. 

Studies performed by Passoni et al. [40] observed a high degree of toxicity in the crude aqueous extract obtained from the leaves of *T. diversifolia*, against Wistar rats, from repeated doses of the extract of the species. However, biological losses were observed only at high doses, above 100 mg·mL^−1^.

### 3.4. Antimicrobial Activity

The results of this research corroborate with data verified by Linthoingambi and Mutum [41], which report the use of organic extracts (petroleum ether, chloroform, and methanol) obtained from the leaves of *T. diversifolia*, perceiving the inhibitory action of growth against bacteria *Enterococcus faecalis*, *Escherichia coli*, *Pseudomonas aeruginosa* and *Staphylococcus aureus*.

The bacterium *E. coli* (ATCC 8739), although part of the intestinal tract, stands out as being widely used for antimicrobial formulation.

However, beings with low immunity make it possible to increase the concentration, which in turn leads to an infection. To this end, the continued use of products known for this treatment has generated mutations and increased resistance of the bacteria. Thus, essential oils with bactericidal potential may be a promising alternative for this problem. In this case, antibiotic treatment becomes necessary [42,43,44,45]. 

For Vahdani et al. [46] essential oils that have piperitone in its composition have bacterial action, which would justify the strong inhibitory action exhibited by the essential oil of *T. diversifolia*, since the substance was the most abundant in the chemical composition of the oil, presenting a percentage of 11.72%.

Odeyemi et al. [47] emphasize the use of *T. diversifolia* crude extracts in methanol, ethanol, and water as promising antibacterial agents since they had antimicrobial effects on *P. aeruginosa*, *Shigella spp*, *Enterococcus spp*, *E. coli* and *Salmonella spp*. However, few studies have described the action of essential oils of *T. diversifolia* against pathogens.

For Guinoiseau et al. [48], the use of essential oils may be more efficient as antimicrobial substances, because their characteristics contribute to the weakening of the cell wall of the bacterium, which could potentiate the antimicrobial activity of *T. diversifolia* species.

## 4. Material and Methods

### 4.1. Collection of Plant Material

The species was collected in the district of Fazendinha-AP, under the coordinates 0°1’17.59” S and 51°6’17.59” W, and later it was sent to the Herbarium of the Institute of Scientific and Technological Research of the State of Amapá—IEPA for the procedures of taxonomic identification and elaboration of exsicatas (sample of pressed species). This was deposited under the number (IAN): 0188.

### 4.2. Obtaining Essential Oil

The essential oil was obtained from the Laboratory of Pharmacognosy and Phytochemistry of the Federal University of Amapá—UNIFAP. The essential oil was obtained by hydrodistillation (temperature 100 °C) in a Clevenger type apparatus for 4 h [49].

### 4.3. Identification of the Chemical Composition of the Essential Oil by Gas Chromatography Coupled to Mass Spectrometry (GC-MS) and Magnetic Nuclear Resonance (NMR)

The analysis of the essential oil was carried out by Gas Chromatography coupled to the Mass Spectrometer (GC-MS) of the Federal University of São Carlos—UFSCar. The Shimadzu equipment, CGEM-SHIMADZU QP 5000, was used. A molten silica capillary column (OPTIMA®-5-0.25 μm), 30 m long and 0.25 mm internal diameter and nitrogen as carrier gas, was used. The operating conditions of the gas chromatograph were: Internal column pressure 67.5 kPa, division ratio 1:20, the gas flow in the column 1.2 mL/min. (210 °C), injector temperature 260 °C, temperature detector or interface (GC-MS) of 280 °C. The initial column temperature was 50 °C, followed by an increase of 6 °C/min. up to 260 °C, this was held constant for 30 min. The mass spectrometer was programmed to perform readings at intervals of 29–400 Da, 0.5 s with ionization energy of 70 eV.

The identification of the chemical compounds present in the essential oil was made from the comparisons of the Linear Indices of Retention (LRI) and Kovats (KI) of the homologous series of n-alkanes (C8–C26) and the literature [15,16]. In addition to the identification made by combining the spectra obtained by the analysis performed on the equipment and the mass spectra of the software library Labsolutions GC-MS solution version 2.50 SU1. 

The structural identification through Nuclear Magnetic Resonance Spectroscopy of Hydrogen 1 and Carbon 13 was developed in the NMR Laboratory of the Department of Chemistry of the Federal University of Amazonas after separation and/or purification of the secondary metabolites of the plant species that compose the essential oils when required.

According to Tavares and Ferreira [50], 4–5 mL of Deuterated Chloroform (CDCl_3_) was added in an aliquot of 0.6 mL of the sample. It was inserted to obtain the fids while processing the spectra in a 9.4 T (400 MHz to H), Bruker brand, model DRX400, with a temperature of 300 K, and a 5 mm reverse detection probe. Hydrogen (^1^H) and carbon (^13^C) spectra was also used.

### 4.4. Analysis of Antioxidant Activity

The tests performed to evaluate the antioxidant activity (% AA) were carried out at the Laboratory of Pharmacognosy and Phytochemistry of the Federal University of Amapá—UNIFAP. According to the methodology of Sousa et al. [25], Lopes-Lutz et al. [51] and Andrade et al. [52], with adaptations; on the consumption of DPPH. 

A methanolic solution of 40 μg·mL^−1^ DPPH was prepared. The essential oil was diluted in methanol at concentrations of 5; 2.5; 1.0; 0.75; 0.50 e 0.25 mg·mL^−1^. For the evaluation, 2.7 mL of DPPH stock solution was added in a test tube, followed by the addition of 0.3 mL of the essential oil solution. The preparation of the negative control was performed from a mixture of 2.7 mL of methanol and the methanolic solution of *T. diversifolia* essential oil. After 30 minutes, spectrophotometric readings (Biospectro SP-22) were performed at 517 nm wavelength [53].

The test was performed in triplicate. The antioxidant activity was calculated according to Gulle et al. [54]: (%AA) = 100 − {[(Abs_sample_ − Abs_white_) × 100] / Abs_control_}%AA = percentage of antioxidant activityAbs_sample_ = Sample AbsorbanceAbs_white_ = Absorbance of whiteAbs_control_ = Control Absorbance

### 4.5. Cytotoxicity Test

The cytotoxicity tests against larvae of *A. salina* Leach were carried out in the Laboratory of Pharmacognosy and Phytochemistry of the Federal University of Amapá—UNIFAP. Using the technique of Araújo et al. [55] and Lôbo et al. [56], with adaptations. First, 250 mL of the 35.5 g·L^−1^ solutions of synthetic sea salt were prepared with 25 mg of *A. salina* eggs exposed to artificial light within 24 h for the hatching of the lavas (methanuplios). Subsequently, these were separated and left in a dark environment, at rest for 24 h to reach the nauplion stage. 

Performing the in vitro tests, a stock solution containing 0.06 g of the essential oil, 28.5 mL of the solution of synthetic sea salt and 1.5 mL of dimethylsulfoxide (DMSO) was added. The nauplii were selected and divided into 7 groups with 10 individuals in each test tube. In each group, an aliquot of stock solution was added and the volume was filled to 5 mL with a solution of synthetic sea salt to obtain final solutions with the following concentrations: 1250; 1000; 500; 250; 100; 50 e 10 μg·mL^−1^. Thus, the groups were assigned according to their concentration and all tests were performed in triplicates. In the end, the death toll *A. saline* was recorded to determine LC_50_ by probit analysis using SPSS software version 22, SPSS Inc., Chicago, IL, USA.

### 4.6. Microbiological Activity 

For the evaluation of the antimicrobial activity, the bacteria *Staphylococcus aureus* (ACTC 6538P), *Escherichia coli* (ATCC 8739) and *Pseudomonas aeruginosa* (ATCC 25922) were used. The results were expressed as Minimum Inhibitory Concentration (MIC) and Minimum Bactericidal Concentration (MBC). MIC was determined by the serial dilution technique in 96-well plates, according to the methodology described in article M07-A10 of 38 manuals [57], with adaptations. To obtain the stock solution the polar solvent DMSO was used to aid in the dilution of the essential oil in distilled water. After this step, 50 μL of the *T. diversifolia* essential oil was added to the first well, with a concentration of 200 mg·mL^−1^.

A serial dilution of well A1 to well A12 was performed. Then, 50 μL of the 0.5 turbid inoculum was added to each well, which on the McFarland nephelometric scale is equivalent to 1.5 × 108 CFU/mL in Mueller-Hinton culture medium. The microplates were placed in an oven for 24 h at 35 ± 2 °C.

As a positive control, amoxicillin (50 mg·mL^−1^) was used; for the negative control, a 4% DMSO solution was used to control the culture environment, bacterial growth and turbidity of the essential oil.

The reading was performed on an Elisa microplate reader (Polaris®), after 24 h with absorbances measured at 630 nm. The values obtained were considered for the creation of graphs that express the viability of the microorganisms followed by statistical analysis performed by analysis of variance (ANOVA) with a confidence interval of 99.9%. Significant differences between averages were determined by the Bonferroni test.

For the determination of Minimum Bactericidal Concentration (MBC), Petri dishes containing Mueller-Hinton agar environment were used with the aid of a sterile bacteriological loop, which was inoculated with 10 μL of the suspension contained in the wells that showed no visual growth during experimental MIC. The plates were housed at 35 °C ± 2, MBC was established as the lowest concentration of test substances capable of completely inhibiting the microbial growth in Petri dishes after 24 h of growth.

### 4.7. Statistical Analysis

Statistical analysis was performed using analysis of variance (ANOVA) with a 95% confidence interval. The significant differences between the means were determined by the Tukey test. 

## 5. Conclusions

The study based on the essential oil of the species *Tithonia diversifolia* allowed the determination of antioxidant, cytotoxic and antimicrobial activities. The main compounds found in the essential oils of this study were *α-*pinene (9.9%), Limonene (5.40%), *(Z)*-*β*-ocimene (4.02%), *p*-cymen-8-ol (3.0%), Piperitone (11.72%), *(E)*-nerolidol (3.78%) and Spathulenol (10.8%). The antioxidant activity proves that the essential oil of this species has antioxidant action, through the DPPH radical capture method, with IC_50_ of 4.30. Its toxicity was elevated with LC_50_ of 3.11 μg·mL^−1^. For the antimicrobial activity, the samples presented promising results against the bacteria *S. aureus*, *E. coli* and *P. aeruginosa* with inhibition of bacterial growth.

## Figures and Tables

**Table 1 pharmaceuticals-12-00034-t001:** Chemical composition of the essential oil of the leaves of *Tithonia diversifolia*.

N° *	LRI	KI	Compounds	Relative Percentage (%)	Identification ^#^
1	945	939	*α-*pinene	9.9	MS, LRI, KI
2	972	975	Sabinene	0.85	MS, IR, KI
3	982	979	*β-*pinene	1.34	MS, IR, KI
4	1028	1029	Limonene	5.40	MS, IR, KI
5	1032	1037	*(Z)-β*-ocimene	4.02	MS, IR, KI
6	1042	1050	*(E)-β-ocimene*	0.09	MS, IR, KI
7	1069	1070	*CIS-*sabinene hydrate	0.3	MS, IR, KI
8	1083	1088	Terpinolene	0.21	MS, IR, KI
9	1090	1091	*p*-cymenene	0.31	MS, IR, KI
10	1099	1096	Linalool	0.32	MS, IR, KI
11	1109	1099	*α-*pinene oxide	0.33	MS, IR, KI
12	1121	1122	*TRANS-*p-mentha-2,8-dien-1-ol	0.35	MS, IR, KI
13	1127	1132	(4*E*,6*Z*)-allo-ocimene	1.3	MS, IR, KI
14	1135	1137	*CIS-*p-mentha-2,8-dien-1-ol	0.39	MS, IR, KI
15	1138	1135	*(Z)-myroxide*	0.25	MS, IR, KI
16	1141	1141	*CIS-*verbenol	0.43	MS, IR, KI
17	1145	1144	*TRANS-*verbenol	1.68	MS, IR, KI
18	1161	1164	*CIS-*Chrysanthenol	0.28	MS, IR, KI
19	1171	1169	Borneol	1.44	MS, IR, KI
20	1180	1177	Terpinen-4-ol	0.31	MS, IR, KI
21	1187	1182	*p*-cymen-8-ol	3.0	MS, IR, KI
22	1195	1193	Dihydro carveol	0.37	MS, IR, KI
23	1207	1205	Verbenone	0.49	MS, IR, KI
24	1218	1216	*TRANS-*carveol	1.0	MS, IR, KI
25	1231	1229	*CIS-*carveol	0.2	MS, IR, KI
26	1243	1243	Carvone	0.42	MS, IR, KI
27	1257	1252	Piperitone	11.72	MS, IR, KI
28	1290	1290	Thymol	0.5	MS, IR, KI
29	1298	1299	Carvacrol	0.57	MS, IR, KI
30	1337	1343	Piperitenone	1.47	MS, IR, KI
31	1349	1389	2-Dodecanone	0.3	MS, IR, KI
32	1411	1426	2,5-dimethoxy-p-cymene	0.83	MS, IR, KI
33	1417	1466	*(E)-*caryophyllene	0.43	MS, IR, KI
34	1477	1488	*(E)-β-i*onone	0.64	MS, IR, KI
35	1489	1493	*α-*zingiberene	1.21	MS, IR, KI
36	1503	1505	*(E,E)-a-*farnesene	0.13	MS, IR, KI
37	1563	1563	*(E)-*nerolidol	3.78	MS, IR, KI
38	1578	1578	Spathulenol	10.8	MS, IR, KI
39	1581	1583	Caryophyllene oxide	3.43	MS, IR, KI
40	1584	1590	Globulol	2.64	MS, IR, KI
41	1624	1631	*β-*muurola-4,10(14)-dienol	1.8	MS, IR, KI
42	1641	1640	epi-*a-*cadinol	2.04	MS, IR, KI
43	1654	1654	*α-*cadinol	1.35	MS, IR, KI
44	1657	1659	Selin-11-en-4-*a*-ol	0.91	MS, IR, KI
45	1663	1671	14-hydroxy-*(Z)-*caryophyllene	0.22	MS, IR, KI
46	1669	1662	Allohimachalol	0.82	MS, IR, KI
47	1683	1671	Bulnesol	0.38	MS, IR, KI
48	1696	1698	(*2Z,6Z)*-farnesol	0.49	MS, IR, KI
49	2106	1943	Phytol	0.78	MS, IR, KI
			Total	80.26	

Notes: * The identification path of the compounds, # identification of the compounds was performed by the mass spectrum (MS) of the library software Labsolutions GC-MS solution version 2.50 SU1 (NIST05 and WILEY’S Library of Mass spectra 9th Edition); Linear Retention Index (LRI) [15] and Kovats Index (KI) [16].

**Table 2 pharmaceuticals-12-00034-t002:** NMR data of the *α*-pinene substance (S 1) compared to literature data.

Position	^1^H NMR	^1^H NMR [17]	^13^C NMR	^13^C NMR [17]
**1**	1.931	1.931	47.05	46.99
**2**	-	-	145.06	144.54
**3**	5.203	5.186	116.01	116.10
**4**	(2.231; 2.210)	2.232	31.26	31.25
**5**	2.065	2.067	40.74	40.69
**6**	-	-	37.96	37.97
**7**	(1.616; 1.559)	(1.15; 2.334)	31.46	31.45
**8**	1.282	1.264	26.35	26.35
**9**	0.853	0.834	20.80	20.80
**10**	1.673	1.659	22.97	23.01

**Table 3 pharmaceuticals-12-00034-t003:** NMR data of the substance Limonene (S 4) compared to literature data.

Position	^1^H NMR	^1^H NMR [18]	^13^C NMR	^13^C NMR [18]
**1**	-	-	133.62	133.30
**2**	2.081	2.082	27.91	27.90
**3**	(1.673; 1.495)	(1.675; 1.495)	30.59	30.60
**4**	1.673	1.675	41.08	41.10
**5**	(2.288; 2.081)	(2.289; 2.082)	30.80	30.80
**6**	5.455	5.209	120.64	120.70
**7**	1.791	1.516	23.46	23.30
**8**	-	-	159.27	149.70
**9**	1.712	1.558	20.67	20.60
**10**	4.770	(4.952; 4.949)	108.35	108.40

**Table 4 pharmaceuticals-12-00034-t004:** NMR data of the substance *(Z)-β*-ocimene (S 5) compared to data in the literature.

Position	^1^H NMR	^1^H NMR [19]	^13^C NMR	^13^C NMR [19]
**1**	(4.995; 5.234)	(4.998; 5.233)	112.57	112.1
**2**	6.393	6.395	141.16	137.5
**3**	-	-	133.74	133.8
**4**	5.368	5.367	133.25	133.1
**5**	2.471	2.528	27.21	27.2
**6**	5.191	5.189	122.3	122.1
**7**	-	-	132.02	132.0
**8**	1.582	1.58	21.81	21.8
**9**	1.749	1.754	14.92	15.1
**10**	1.569	1.58	21.81	21.8

**Table 5 pharmaceuticals-12-00034-t005:** NMR data of *p*-cymen-8-ol substance (S 21) compared to literature data.

Position	^1^H NMR	^1^H NMR [19]	^13^C NMR	^13^C NMR [19]
**1**	-	-	141.16	139.8
**2**	7.160	7.104	128.87	128.7
**3**	7.243	7.243	124.30	124.3
**4**	-	-	150.27	146.3
**5**	7.243	7.243	124.30	124.3
**6**	7.160	7.104	128.87	128.7
**7**	2.252	2.252	21.41	21.3
**8**	-	-	70.49	71.4
**9**	1.382	1.381	31.74	31.8
**10**	1.382	1.381	31.74	31.8

**Table 6 pharmaceuticals-12-00034-t006:** NMR data of Piperitone substance (S 27) compared to literature data.

Position	^1^H NMR	^1^H NMR [20]	^13^C NMR	^13^C NMR [20]
**1**	-	-	201.32	200.0
**2**	5.963	5.951	126.83	126.8
**3**		-	161.08	161.6
**4**	(2.296; 2.312)	(2.298; 2.309)	30.59	30.5
**5**	(1.911; 1.846)	(1.910; 1.843)	22.97	23.2
**6**	2.679	2.679	51.58	51.6
**7**	2.186	2.179	24.06	24.1
**8**	1.911	1.910	25.83	25.9
**9**	0.952	0.956	20.26	20.1
**10**	0.952	0.956	20.18	20.1

**Table 7 pharmaceuticals-12-00034-t007:** NMR data of *(E)-*nerolidol (S 37) substance compared to literature data.

Position	^1^H NMR	^1^H NMR [21]	^13^C NMR	^13^C NMR [21]
**1**	(5.087; 5.017)	(5.085; 5.016)	111.65	111.54
**2**	5.860	5.878	144.52	144.86
**3**	-	-	70.49	73.01
**4**	(1.382; 1.623)	(1.388; 1.602)	41.74	41.91
**5**	2.034	2.033	22.61	22.61
**6**	5.277	5.283	124.25	124.09
**7**	-	-	133.74	134.63
**8**	1.894	1.894	39.65	39.46
**9**	2.229	2.200	26.45	26.41
**10**	5.287	5.288	124.25	124.13
**11**	-	-	130.66	130.79
**12**	1.530	1.538	25.66	25.55
**13**	1.530	1.538	16.98	17.33
**14**	1.604	1.602	15.88	15.66
**15**	1.382	1.388	27.37	27.31

**Table 8 pharmaceuticals-12-00034-t008:** NMR data of Spathulenol substance (S 38) compared to literature data.

Position	^1^H NMR	^1^H NMR [22]	^13^C NMR	^13^C NMR [22]
**1**	2.384	2.387	53.40	53.37
**2**	(1.996; 1.681)	(1.996; 1.686)	26.67	26.68
**3**	(1.530; 1.771)	(1.538; 1.778)	41.74	41.71
**4**	-	-	80.99	80.90
**5**	1.530	1.538	54.33	54.27
**6**	0.853	0.837	29.92	29.90
**7**	0.912	0.912	27.46	27.46
**8**	(1.530; 1.521)	(1.538; 1.522)	24.78	24.74
**9**	(2.380; 2.384)	(2.381; 2.387)	38.86	38.83
**10**	-	-	153.43	153.38
**11**	-	-	20.18	20.21
**12**	1.141	1.143	28.65	28.62
**13**	1.141	1.143	16.32	16.29
**14**	(4.875; 5.065)	(4.861; 5.061)	28.7	28.65
**15**	1.327	1.329	106.25	106.22

**Table 9 pharmaceuticals-12-00034-t009:** NMR data of Caryophyllene oxide (S 39) compared to literature data.

Position	^1^H NMR	^1^H NMR [23]	^13^C NMR	^13^C NMR [23]
**1**	1.73	1.76	50.7	50.9
**2**	(1.57; 1.63)	(1.45; 1.63)	27.9	27.2
**3**	(0.94; 2.03)	(0.95; 2,06)	39.1	39.2
**4**	-	-	54.3	59.6
**5**	2.85	2.86	6.37	63.6
**6**	(1.28; 2.29)	(1.28; 2.23)	30.1	30.1
**7**	(2.16; 2.85)	(2.11; 2.37)	29.8	29.8
**8**	-	-	150.2	151.7
**9**	(2.85)	2.60	48.7	48.7
**10**	(1.43; 1.48)	(1.43; 1.47)	39.7	39.8
**11**	-	-	33.5	33.9
**12**	(4.80; 4.97)	(4.81; 4.99)	16.9	16.9
**13**	1.20	1.19	112.7	112.7
**14**	(0.98; 1.00)	(0.98; 1.01)	29.8	29.8
**15**	(0.98; 1.00)	(0.98; 1.01)	20.8	21.6

**Table 10 pharmaceuticals-12-00034-t010:** %AA values of the essential oil of the leaves of *T. diversifolia*.

	Concentration (mg·mL^−1^)						
**Plant species**	5	2.5	1	0.75	0.5	0.25	IC_50_
***T. diversifolia***	54.6 ± 0.06 ^a^	37.4 ± 0.27 ^b^	27.4 ± 0.41 ^c^	23.9 ± 0.55 ^d^	21.8 ± 0.13 ^eg^	20.2 ± 0.79 ^fg^	4.30

**Table 11 pharmaceuticals-12-00034-t011:** % AA values of the Ascorbic acid.

	Concentration (µg·mL^−1^)						
**compound**	250	125	62.5	31.25	15.62	7.81	IC_50_
***Ascorbic acid***	99.99 ± 0.0	99.99 ± 0.0	99.99 ± 0.0	99.93 ± 0.02	30 ± 0.10	18.57 ± 0.52	16.71

**Table 12 pharmaceuticals-12-00034-t012:** Mortality of *A. salina* larvae at different concentrations of *T. diversifolia* essential oil.

	Concentration (µg·mL^−1^)							
**Plant species**	1250	1000	500	250	100	50	10	LC_50_
***T. diversifolia***	100 ^a^	100 ^a^	100 ^a^	100 ^a^	100 ^a^	98.1 ^b^	83.5 ^c^	3.11

**Table 13 pharmaceuticals-12-00034-t013:** Minimum Inhibitory Concentration (MIC) of OE *T. diversifolia*.

Plant Species *T. diversifolia*				
**Bacterium**	MIC (mg·mL^−1^)			
	100	50	25	12.5
*Staphylococcus aureus*	+	+	NA	NA
*Escherichia coli*	+	+	+	+
*Pseudomonas aeruginosa*	+	+	NA	NA
Amoxicillin (Positive Control)	+	+	+	+
DMSO (Negative control)	NA	NA	NA	NA

NA: it did not show.

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
