# Peer review of "Chemical Characterization, Antioxidant, Cytotoxic and Microbiological Activities of the Essential Oil of Leaf of Tithonia Diversifolia (Hemsl) A. Gray (Asteraceae)"

_pharmaceuticals, 2019, doi:10.3390/ph12010034_

Reviewer 1 Report

In this manuscript entitled “Chemical characterization, antioxidant, cytotoxic and microbiological activities of the essential oil of leaf of Tithonia diversifolia (Hemsl) A. Gray (Asteraceae),” the authors determined antioxidant, cytotoxic and microbiological activities of the essential oil of leaf of Tithonia diversifolia (Hemsl). The experiments were well conducted in this manuscript. My brief comments are as follows:

Comments:

1.    In the abstract, the authors used E. coli to evaluate the antimicrobial activities. Is it usual even though it is not pathogenic bacteria?

2.    In page 6, line 73, 2.2. antioxidant activity. There are numerous methods to evaluate antioxidant activity. The authors should describe the principal of used method in the present study, here.

3.    In page 6, line 84, 2.3. cytotoxic test. The authors used nauplii as a stressor to induce cytotoxicity. Please describe the biological and/or pharmaceutical meanings to use this method.

4.    In page 7, line 124, 3.2. antioxidant activity. To evaluate the significance of the antioxidative activity, the comparison with known antioxidants is important. The authors should compare your compounds with other known antioxidants such as Trolox to evaluate the significance of these compounds.

Author Response

Reviewer 01: in his comments, the reviewer made four (4) questions, the answers are as follows: 

1.  In the abstract, the authors used E. coli to evaluate the antimicrobial activities. Is it usual even though it is not pathogenic bacteria?

Answer: The bacterium Escherichia coli (ATCC 8739) is part of the intestinal tract. However, beings with low immunity make it possible to increase the concentration, which in turn leads to an infection. To this end, the continued use of products known for this treatment has generated mutations and increased resistance of bacteria [40-42]. In this case, essential oils with bactericidal potential may be a promising alternative for this problem. In that case, antibiotic treatment becomes necessary. In addition, the American Type Culture Collection manual [43] emphasizes that E. coli is used for antimicrobial formulation.

2. In page 6, line 73, 2.2. antioxidant activity. There are numerous methods to evaluate antioxidant activity. The authors should describe the principal of used method in the present study, here.

Answer: The antioxidant activity was performed by the DPPH radical capture method (or DPPH consumption), this methodology consists in the reading of absorbance through a Spectrophotometer, the higher these values of absorbances, the greater the antioxidant activity. The determination of the antioxidant activity was performed using the DPPH free radical capture method, however, it is known that the essential oils are complex mixtures and it is difficult to attribute certain biological activities. In the present assay, the elimination of the radical was shown to be low. Presented an aspect still slightly purple. The antioxidant activity is directly related to the free radical consumption, the higher this consumption, the greater the activity and the lower the inhibitory concentration, in this case the IC 50 [23-25].

3. In page 6, line 84, 2.3. cytotoxic test. The authors used nauplii as a stressor to induce cytotoxicity. Please describe the biological and/or pharmaceutical meanings to use this method.

Answer:  larvae of A. salina (nauplii), are effective to verify if a certain plant sample is toxic.  Preliminary toxicity testing on plant species is performed to verify that the study material may contain substances that may cause harm to the health of a population. In addition, toxicity results may suggest that the test material (T. diversifolia essential oil) may be used for biocidal, repellent and antitumor activities [31-33]. Being that, it is a preliminary test, with low cost and selective for tests of more specific toxicities.

4. In page 7, line 124, 3.2. antioxidant activity. To evaluate the significance of the antioxidative activity, the comparison with known antioxidants is important. The authors should compare your compounds with other known antioxidants such as Trolox to evaluate the significance of these compounds.

Answer: To evaluate the antioxidant potential the essential oil of T. diversifolia was used as an antioxidant standard constituent ascorbic acid. For this, an assay was performed using the DPPH free radical consumption method. The values obtained indicated that compared with this standard (ascorbic acid), the essential oil of T. diversifolia had low antioxidant activity. The IC50 of ascorbic acid presented an IC50 of 16.71 μg.mL-1 (Table 10), a much lower value than the essential oil, which was 4.30 mg.mL-1.

Table 10: %AA of ascorbic acid.

Concentration (µg.mL-1)

compound

250

125

62.5

31.25

15.62

7.81

IC50

Ascorbic acid

99.99

±0.0

99.99

±0.0

99.99

±0.0

99.93

±0,02

30

±0.10

18.57

±0.52

16.71

References (All references used to answer the previous questions were inserted in the references of the manuscript)

40.       Emirdağ-Öztürk, S.; Karayildirim, T.; Anil, H. Synthesis of egonol derivatives and their antimicrobial activities. Bioorganic & Medicinal Chemistry 2011, 19, 1179–1188.

41.       Nagai, N.; Yoshioka, C.; Mano, Y.; Tnabe, W.; Ito, Y.; Okamoto, N.; Shimomura, Y. A nanoparticle formulation of disulfiram prolongs corneal residence time of the drug and reduces intraocular pressure. Experimental Eye Research 2015, 132, 115–123.

42.       Ateufack, G.; Nana Yousseu, W.; Dongmo Feudjio, B.; Fonkeng Sama, L.; kuiate, J.; Kamanyi, A. Antidiarrheal and in vitro antibacterial activities of leaves extracts of Hibiscus asper. Hook. F. (Malvaceae). Asian J Pharm Clin Res 2014, 7, 130–136.

43.       American Type Culture Collection manuscript in the following manner: Escherichia coli (ATCC® 8739TM) 2018.

23.       Sousa, C.M. de M.; Silva, H.R. e; Vieira-Jr., G.M.; Ayres, M.C.C.; Costa, C.L.S. da; Araújo, D.S.; Cavalcante, L.C.D.; Barros, E.D.S.; Araújo, P.B. de M.; Brandão, M.S.; et al. Fenóis totais e atividade antioxidante de cinco plantas medicinais. Química Nova 2007, 30, 351–355.

24.       Bondet, V.; Brand-Williams, W.; Berset, C. Kinetics and Mechanisms of Antioxidant Activity using the DPPH.Free Radical Method. LWT - Food Science and Technology 1997, 30, 609–615.

25.       Zhang, H.-Y.; Gao, Y.; Lai, P.-X. Chemical Composition, Antioxidant, Antimicrobial and Cytotoxic Activities of Essential Oil from Premna microphylla Turczaninow. Molecules 2017, 22, 381.

31.       Hyacienth, D.C.; Almeida, S.S.M.S. Estudo Fitoquímico, Toxicidade em Artemia salina Leach e Atividade Antibacteriana de Pseudoxandra cuspidata Maas. Biota Amazônia 2015, 5, 4–7.

32.       Candido, L.P. Busca de extratos e compostos ativos com potencial herbicida e inseticida nas espécies Davilla elliptica St. Hill e Ocotea pulchella Nees & Mart. Tese, UNIVERSIDADE FEDERAL DE SÃO CARLOS: São Carlos, 2016.

33.       Li, X.; Huang, G.; Zhao, G.; Chen, W.; Li, J.; Sun, L. Two New Monoterpenes from Tithonia diversifolia and Their Anti-Hyperglycemic Activity. Rec. Nat. Prod. 2013,7  351-354

Reviewer 2 Report

Reviewer

ABSTRACT:

Lines 16, 17: are not present symbols (a…)

Line 17: Z, E,  italic

INTRODUCTION:

Line 29: family names must be into italic

Line 48. Idem

RESULTS

Lines 52-54  Please use the correct way for chemical constituents

for example

Pinene  is                    a-pinene

Cymen-8-ol    is                p-cymen-8-ol

Table 1

First line is to change into English

All the constituents are to be write lowercase character preceded by symbol, the position m, p, o and also by E or Z (or CIS or TRANS) italic

You have to add a column with the indication of the way of identification of compounds: MS-RI, Std, MS ?????

You have to add a column for reference of particular constituents

In my opinion the chemical characterization of the essential oil can be carried out with more accurate GC/MS analyses

FIGURES

Are all well-known constituents of essential oils they are unnecessary

NMR

It is not clear whether the authors compared the NMR spectra also  with their own database or only with literature spectra. In the last case it is very difficult to assign with certainty the signal belonging to a specific compound. Many publications are available on the characterization of the essential oil using NMR, but comparing the new oil with the self-built database (see for example many articles by Casanova J. et al. of the University of Corsica). If are compared only with literature data and not with pure standards it does not make sense

In this view Discussion on Chemical characterization of essential oil it would be useless

In my opinion the paper need major revision to be accepted.            

Author Response

Reviewer 02:

ABSTRACT:

As requested by the reviewer the lines16, 17 that did not include the (α) symbol was already run in the manuscript.

In line 17 the symbols Z, E were changed to italic formatting.

INTRODUCTION:

In lines 29 and 48: the names of the families have already been changed to italics in the manuscript.

RESULTS

In lines 52-54 the change in the formatting of the nomenclature of chemical constituents was requested. This request was followed and corrections were made.

The first line of table 1 has been translated into English as per request.

The reviewer requested that all constituents should be written in lowercase characters preceded by a symbol in position m, p, o and also by E or Z (or CIS or TRANS) that should be in italics. These guidelines were followed and all corrections were made in the manuscript.

It was suggested by the broker that he add a column indicating the path of identification of the compounds. This request was answered and inserted in the manuscript.

FIGURES 1

It was placed by the reviewer that there is no need for a figure with all known constituents of the essential oils. In this case, the request was accepted, and this figure was removed from the manuscript.

In relation to GC-MS, the requested modifications were made. In addition, the mass spectra data of each major compound found in T. diversifolia essential oil analysis was inserted into Supplementary Material 1. GC-MS.

NMR

The NMR analysis was performed with the essential oil of T. diversifolia and in comparison with literature data, it was not compared with its own database and nor were standards used.

In this way, the discussion of NMR was withdrawn, as suggested by the reviewer.

Round  2

Reviewer 2 Report

The paper is acepted